# Multi-Image Encryption Algorithm Based on Cascaded Modulation Chaotic System and Block-Scrambling-Diffusion

**DOI:** 10.3390/e24081053

**Published:** 2022-07-31

**Authors:** Ting Wang, Bin Ge, Chenxing Xia, Gaole Dai

**Affiliations:** College of Computer Science and Engineering, Anhui University of Science and Technology, Huainan 232001, China; 2020200993@aust.edu.cn (T.W.); cxxia@aust.edu.cn (C.X.); 2020201101@aust.edu.cn (G.D.)

**Keywords:** cascade modulation chaotic system, multi-image encryption, block-scrambling-diffusion, double scrambling mechanism, bit-group diffusion

## Abstract

To address the problem of a poor security image encryption algorithm based on a single chaotic map, this paper proposes a cascade modulation chaotic system (CMCS) that can generate multiple chaotic maps. On this basis, a multi-image encryption algorithm with block-scrambling-diffusion is proposed using CMCS. The algorithm makes full use of the features of CMCS to achieve the effect of one encryption at a time for images. Firstly, the key-value associated with the plaintexts is generated using a secure hash algorithm-512 (SHA-512) operation and random sequence, and the three images are fully confused by the double scrambling mechanism. Secondly, the scrambled image is converted into a bit-level matrix, and the pixel values are evenly distributed using the bit-group diffusion. Finally, the non-sequence diffusion of hexadecimal addition and subtraction rules is used to improve the security of the encryption algorithm. Experimental results demonstrate that the encryption algorithm proposed in this paper has a good encryption effect and can resist various attacks.

## 1. Introduction

The rapid and continuous transmission of digital information over the network platform has made network security a common concern. As an essential means of network digital information protection, image encryption converts meaningful original images into encrypted images that are unrecognized and similar to noise to ensure the safe transmission of information in the network [1]. Although traditional encryption technologies such as DES and AES have achieved good results in text encryption, due to a large amount of image data and high computational complexity, the effect of image encryption using DES and AES for text information encryption is not particularly ideal. The randomness and hybridization of chaotic systems are similar to the scrambling and diffusion in cryptography [2]. Therefore, chaotic systems are widely used in the field of image encryption.

Image encryption schemes based on chaotic systems have undergone a series of changes. Earlier chaos-based image encryption algorithms generally used a single chaotic map [3,4]. Due to the development of the times, a single chaotic map can no longer guarantee the secure transmission of images. For this reason, many scholars have proposed an image encryption scheme based on double chaos. For example, Xu et al. [5] converted the image into a bit-level matrix, using the Logistic map for pixel scrambling and the Chen map for pixel diffusion. Kurunandan et al. [6] applied a Cat map and 2D-LSCM map to the medical image encryption by scrambling-diffusion. The use of the multi-chaotic map improves the security of the algorithm.

In addition, early image encryption schemes typically used a single scrambled encryption [7,8]. This kind of encryption scheme only changes the pixel’s position and does not change the statistical characteristics of the pixel, which has great security risks. Current image encryption algorithms are mainly based on the cipher principles of Shannon design, including two parts, scrambling and diffusion. The encryption technology of this structure can be traced back to the symmetric block encryption technology proposed by Fridrich in 1997 [9]. Since then, many image encryption systems based on scrambling-diffusion structures have been developed [10,11]. Chai et al. [10] decomposed the plaintext image into eight planes for scrambling and then performed bidirectional diffusion on the scrambled image to obtain the encrypted image. Ref. [11] proposes a robust image encryption scheme based on chaotic systems and elliptic curves over finite fields. In this scheme, Arnold transform is used to scramble the image to be encrypted, and then the pixel value of the scrambled image is mixed with the random sequence XOR to generate the encrypted image. In References [10,11], the diffusion operation was added, which changed the value of pixels and enhanced the ability of the algorithm to resist differential attacks. On this basis, Talhaoui et al. [12] proposed to combine the scrambling and diffusion processes. Scrambling and diffusion only take one stage, which reduces the time required for encryption and effectively improves the speed of image encryption. However, the above encryption algorithms, which only use a single scrambling-diffusion to encrypt the image, can no longer guarantee high image security. To this end, Zhang et al. [13] used the Arnold transform to perform primary scrambling of the plaintext image. Then the image was scrambled twice using phase truncated fractional Fourier transform and random masks. Finally, DNA diffusion was used to obtain ciphertext images. Chen et al. [14] proposed an image encryption structure based on diffusion- scrambling- diffusion. Through two rounds of diffusion, the small differences in the plaintext image are diffused to all pixels of the encrypted image.

Currently, most image encryption algorithms are based on pixel-level scrambling-diffusion [15,16,17,18,19]. Because bit-level operation can achieve better scrambling and diffusion effect, the bit-level operation is often used in some encryption schemes [20,21,22]. For example, Sujarani et al. [20] proposed a dynamic bit-flip diffusion image encryption algorithm. Zhu et al. [21] proposed a 3D bit-level image encryption scheme using Rubik’s cube method. Li et al. [22] proposed an image encryption scheme combining bit-level scrambling and multiplication diffusion. However, algorithms based on bit-level operations need to process eight times as much data as those based on pixel-based operations. Therefore, improving the speed of the bit-level encryption algorithm becomes particularly important.

Meanwhile, most image encryption algorithms are limited to gray images [23,24,25]. The color image has three planes and higher data redundancy than the gray image. However, whether it is based on color image encryption algorithm [26,27,28] or gray image encryption, the algorithm encrypts a single image, cannot encrypt multiple images simultaneously, and the encryption efficiency is low. As a result, multi-image encryption’s high security and efficiency have become a new demand.

This paper presents a multi-image encryption algorithm based on a cascade modulation system and block scrambling-diffusion. The purpose of this paper is to construct a chaotic system that can generate multiple chaotic maps and improve the randomness of sequences. The simultaneous encryption of three gray images is realized by using the characteristics of image channels; a block scrambling-diffusion mechanism is established to make the image of mutual influence and full confusion, to improve the security of the encryption algorithm. The security of the encryption algorithm can be measured by simulation experiments in Section 5: key space, key sensitivity, histogram, correlation, information entropy and anti-differential attack. At the same time, the anti-cutting attack analysis and anti-noise attack analysis in Section 5 can reflect the robustness of the algorithm. The main innovations and contributions of the algorithm are as follows:(1)Although the image encryption algorithm based on a single chaotic map can achieve a certain encryption effect, the complexity is not high. Therefore, this paper proposes a cascaded modulated chaotic system (CMCS) as the key generation source.(2)To solve the problem that the scrambling and diffusion steps are independent of the plaintext image, the initial value of CMCS and the generation of system parameters depend on the plaintext image, which can effectively brute force and plaintext attacks.(3)In the scrambling process, three gray images are fused into a color image, and the three images are divided into blocks. On the basis, through cross-plane scrambling, the three images influence each other. In addition, the chaotic sequence is used to scramble intra-block to reduce the correlation between adjacent pixels.(4)According to the characteristics of the bit-level matrix, it is divided into four types. The corresponding diffusion algorithm is adopted for the grouping type to make the pixel distribution more average. Meanwhile, the non-sequential diffusion of the hexadecimal addition and subtraction rule makes the non-linear relationship between plaintext and ciphertext more complex. It improves the ability of the algorithm to resist selective plaintext attacks.

The paper is organized as follows: Section 2 briefly introduces four traditional chaotic maps and analyzes the characteristics of the map by using the bifurcation diagram. Section 3 first defines the structure of the cascaded modulated chaotic system (CMCS). Then, it lists three examples of maps generated by CMCS. Finally, it verifies its superior chaos through bifurcation diagrams, Lyapunov exponents, and 0–1 tests. Section 4 details the specific steps of encryption and decryption based on CMCS and block scrambling diffusion. Section 5 gives the image encryption results using Matlab simulation experiments. A summary of this paper is presented in Section 6.

## 2. Background

The simulation results of the chaotic map introduced in this paper are all carried out in the Matlab environment. In Matlab simulation software, the value of π is determined. When using Matlab for calculation, to directly input π, pi is used to represent π, and the last four digits of the π decimal point are retained by rounding. That is, pi = π = 3.1416 in Matlab. At the same time, Matlab uses double precision or single precision format to represent the number of floating points. The highest precision of Matlab is double precision and the default value is double precision, including 16 valid numbers. This can effectively avoid the small errors generated in the calculation due to different systems, to avoid the problem of different keys caused by different calculation accuracy between the encryption and decryption parties under the same initial value. Therefore, the map generated by the system can be used for image encryption.

### 2.1. Henon Map

Henon map is a two-dimensional discrete chaotic map and the simplest non-linear map in high-dimensional map. Its mathematical definition is as follows:(1)$$\left\{\begin{array}{c} x_{n+1}=1+y_{n}-ax_{n}^{2}\\ y_{n+1}=bx_{n} \end{array}\right.$$
where $x_{n}$ and $y_{n}$ are output chaotic sequences, *a* and *b* are system parameters for the Henon map. When *b* = 0.3 and *a* ∈ [1.06, 1.4], the Henon map enters a chaotic state. Its bifurcation diagram is shown in Figure 1a.

### 2.2. Logistic Map

One-dimensional Logistic chaotic system is often used in image encryption algorithms as key generation sources. Its mathematical expression is:(2)$$x_{n+1}=\mu \times x_{n}\times \left(1-x_{n}\right)$$In Equation (Equation 2), μ is the control parameter and μ ∈ (0, 4), $x_{n}$ is chaotic map value and $x_{n}$ ∈ [0, 1]. When the value of μ is determined, the logistic map is sensitive to the initial value. It can be observed from Figure 1b that when μ ∈ (3.57, 4), the system enters a chaotic state. Although Logistic map is simple and efficient, it also has shortcomings such as a small key space and poor performance of chaotic sequences.

### 2.3. Sine Map

The Sine map is derived from sine function as one-dimensional map, which retains the output range of sine function [0, 1], but changes the input range from [0, π] to [0, 1]. The mathematical definition is as follows:(3)$$x_{n+1}=\mu \sin\left(\pi x_{n}\right)$$
where $x_{n}$ is the output chaotic sequence and μ ∈ [0, 1] is the system parameter of the Sine map. From Figure 1c, it can be observed that when μ ∈ [0.87, 1], the Sine map is chaotic and can be used for image encryption. Meanwhile, from Figure 1b,c, we can observe that the chaotic behavior of the Sine map and Logistic map are similar, but the chaotic interval is different.

### 2.4. Iterative Map

The Iterative map and Sine map are related to the sine function. However, the form of Iterative map is more complex. Its mathematical definition is as follows:(4)$$x_{n+1}=\mu \times x_{n}\times \left(1-x_{n}\right)$$
where $x_{n}$ is the output chaotic sequence and *a* ∈ [0, 1] is the system parameter of the Iterative map. When *a* = 0.5, its bifurcation diagram is shown in Figure 1d. From Figure 1d, it can be observed that the Iterative map can reach the full map state within a certain range, although there are many empty window periods. In addition, as shown in Figure 1c,d, the Iterative map has a larger chaotic range compared with the Sine map.

## 3. Chaos System

It can be observed from Figure 1a–d that the traditional one-dimensional map and two-dimensional map have certain chaos under appropriate parameters. However, their chaotic interval is too small and the chaotic behavior is relatively simple, which leads to the encryption effect not being ideal. For this reason, this paper proposes the cascaded modulation chaotic system (CMCS) and evaluates the performance of the chaotic system through three examples.

### 3.1. Definition of Chaotic System

CMCS uses a two-dimensional chaotic map and a one-dimensional chaotic map (called seed map) to generate a large number of new two-dimensional chaotic maps. Assume that *f*(*x*) is a linear function, *F*(*x*) and *G*(*x*) are chaotic maps. CMCS is defined as follows:(5)$$\left\{\begin{array}{c} x_{n+1}=F\left(f\left(x_{n}\right)\cdot G\left(x_{n}\right)\right)\\ y_{n+1}=F\left(f\left(y_{n}\right)\cdot G\left(y_{n}\right)\right) \end{array}\right.$$
where $x_{n}$ and $y_{n}$ are output chaotic sequences. In CMCS, the linear function *f*(*x*) modulates the chaotic map *G*(*x*). Then, the modulation result is used as the input of the chaotic map *F*(*x*) further to improve the dynamic characteristics of the chaotic system. At the same time, using the cascade method to make chaotic maps interact with each other can produce more complex chaotic sequences.

### 3.2. Examples of the Proposed Chaotic System

This section lists three examples and analyzes the chaotic characteristics of the example by bifurcation diagram (BD), Lyapunov exponent (LE) and 0–1 test. Experimental analysis shows that the newly generated chaotic map has a more extensive chaotic range, less empty window period and more complex chaotic behavior than the original seed map.

#### 3.2.1. Logistic-Henon Cascade Map (LHCM)

*F*(*x*) selects a two-dimensional Henon map and *G*(*x*) selects a one-dimensional Logistic map. The Logistic map is appropriately scaled by *f*(*x*) = *x* + 1, and then the Logistic-Henon cascade map (LHCM) is obtained by the cascade method as follows:(6)$$\left\{\begin{array}{c} x_{n+1}=y_{n}+1-a{\left(x\left(1-x\right)\left(1+x\right)\right)}^{2}\\ y_{n+1}=rbx\left(1-x\right)\left(1+y\right) \end{array}\right.$$
where *a* ∈ [1, 1.6], *r* and *b* are the control parameter of LHCM map, and $x_{n}$ and $y_{n}$ are the output chaotic sequence. When *a* ∈ [1, 1.6], *r* = 4 and *b* = 0.3, the BD, the LE and 0–1 test results of LHCM are shown in Figure 2a–c.

First, Figure 2a shows the BD of LHCM. From Figure 2a, we can see that the LHCM map can reach the state of the whole map under certain parameters. Compared with Figure 1a,b, the chaotic interval of LHCM is larger. Second, having more than one positive LE value is necessary for a dynamic system to have chaotic behavior. The LE of LHCM is shown in Figure 2b, and the one LE value of the LHCM map is positive. It shows that LHCM is chaotic in a certain parameter range. Finally, Figure 2c shows the 0–1 test results of LHCM. The output of the 0–1 test is close to 1 indicating sequence chaos and close to 0 indicating sequence regularity. Figure 2c shows that the 0–1 test output of the LHCM map is close to 1, that is, the sequence chaos generated by the LHCM map. Compared with Logistic and Henon, the chaotic sequence generated by LHCM is more random and more suitable for image encryption.

#### 3.2.2. Henon-Sine Cascade Map (HSCM)

*F*(*x*) is chosen as a one-dimensional Sine map and *G*(*x*) is chosen as a two-dimensional Henon map. Let *f*(*x*) = *x*, *x* be used to modulate the Henon map and concatenate the modulation results with the Sine map. Finally, the Sine map is extended from one-dimensional to two-dimensional so that the output is intertwined to obtain the Henon-Sine cascade map (HSCM), as follows:(7)$$\left\{\begin{array}{c} x_{n+1}=r\sin\left(\pi x_{n}\left(1-ax_{n}^{2}+y_{n}\right)\right)\\ y_{n+1}=r\sin\left(\pi y_{n}bx_{n}\right) \end{array}\right.$$
where *r*, *a* and *b* are the control parameters of the HSCM map. Randomly select the initial values $x_{0}$ ∈ (0, 1] and $y_{0}$ ∈ (0, 1], let *b* = 0.3, *r* = 4, the BD, LE and 0–1 test are shown in Figure 2d–f. From Figure 2d, it can be observed that the chaotic sequence generated by HSCM is uniformly distributed in [0, 1]. Within the range of parameters, HSCM almost has no empty window period. HSCM has better chaotic characteristics than the seed map. From Figure 2e, it can be observed that the HSCM map has two positive LE values, which is a hyperchaotic map. From Figure 2, it is known that the output result of the HSCM map is close to 1, which indicates that the LHCM map has good chaotic characteristics.

#### 3.2.3. Henon-Iterative Cascade Map (HICM)

*F*(*x*) selects a one-dimensional Iterative map, *G*(*x*) selects a two-dimensional Henon map, let *f*(*x*) = *x* to obtain the Henon- Iterative cascade map (HICM) as follows:(8)$$\left\{\begin{array}{c} x_{n+1}=\sin\left(r\pi /\left(x_{n}\left(1-ax_{n}^{2}+y_{n}\right)\right)\right)\\ y_{n+1}=\sin\left(r\pi /\left(bx_{n}y_{n}\right)\right) \end{array}\right.$$
where r>0, a>0 and b>0 are the control parameters of the HICM map and $x_{n}$, $y_{n}$ are the output chaotic sequences.

Figure 2g–i describes the BD, LE and 0–1 test of HICM for states $x_{0}$ ∈ (0, 1], $y_{0}$ ∈ (0, 1], *b* = 0.3, *r* = 4. It can be observed from Figure 2g that no matter what value a takes, HICM enters a chaotic state, which is a complete graph. Compared with Equations (1) and (4), HICM increases the key space and the range of a. Moreover, Figure 2h,i shows that HICM has two positive LE values, and the output of the 0–1 test is close to 1. HICM has a good chaotic behavior similar to LHCM and HSCM.

In summary, it can be observed from Figure 2 that under certain conditions, the bifurcation diagrams of LHCM, HSCM, and HICM show random distribution, and their trajectory is difficult to predict, which can be used for the generation of random sequences. The Lyapunov exponent is a quantitative description of the sensitivity of chaotic systems to small changes in initial conditions. When the system has a positive LE value, the system is chaotic. LHCM, HSCM, and HICM maps have positive LE values, indicating that they have initial sensitivity, which coincides with the key sensitivity required in the encryption process. Because LHCM, HSCM, and HICM map trajectories are difficult to predict and sensitive to the initial value, when the map is used for image encryption, it is difficult for attackers to speculate its equation, which can improve the security of the algorithm to a certain extent.

## 4. Encryption and Decryption Algorithm

To improve the efficiency and security of the encryption algorithm, we designed a CMCS-based multi-image encryption scheme in this section. In the scrambling-diffusion stage, the random sequence generated by different chaotic maps can effectively improve the security of the encryption algorithm. At the same time, the double scrambling mechanism between inter-block and intra-block, as well as the multiple diffusion of bit-level and hexadecimal, can break the strong correlation between pixels and make pixel values uniformly distributed to resist various attacks. The whole process of the encryption algorithm is shown in Figure 3.

### 4.1. Generating the Initial Value Key

The scrambling-diffusion process of many algorithms is independent of plaintext images. It leads to the algorithm not being sensitive to plaintext images and keys and cannot resist known-plaintext attacks or differential attacks. Refs. [29,30,31] shows some typical methods for cracking cryptosystems based on scrambling-diffusion structures. To solve the problems of low-density sensitivity and plaintext sensitivity, researchers have proposed some image encryption schemes related to plaintext in recent years [32,33]. They all relate the key system to the characteristics of the plaintext image through different methods.

SHA-512 is used to process the plaintext image to generate 128-bit hexadecimal digest values and convert them into 512-bit binary numbers to realize the key associated with the plaintext. Assuming that the generated first group of digest values is $k_{1}$, the second group of digest values is $k_{2}$, the third group of digest values is $k_{3}$. The initial key *k* of 512 bits is obtained by XOR between three groups of summary values and 512 bits random sequence $k_{4}$. Then $k=\left\{x_{0},y_{0},r,b,w_{1},w_{2},w_{3},w_{4}\right\}$ can be obtained from Equation (Equation 9).
(9)$$k=k_{1}⊕k_{2}⊕k_{3}⊕k_{4}$$
where $x_{0}$, $y_{0}$, *r* and *b* are the initial value and control parameters of LHCM, HSCM and HICM. $w_{i}$ is the interference parameter used to change the value of the control parameters.

The initial value of the chaotic map is generated by the key *k*, which can achieve the effect of a one-time-one-key. The initial state of the chaotic map is generated as shown in Algorithm 1. Where $sum_{1}$ is the sum of the pixel values of the first gray image, $sum_{2}$ is the sum of the pixel values of the second gray image, $sum_{3}$ is the sum of the pixel values of the third gray image, sum is the sum of the pixel values of the three gray images.
**Algorithm 1** Generate the initial values and control parameters of the chaotic mapInput: key *k* with length of 512 bits.Output: Initial state ($x_{0}^{1},y_{0}^{1},r_{1},b_{1}$), ($x_{0}^{2},y_{0}^{2},r_{2},b_{2}$), ($x_{0}^{3},y_{0}^{3},r_{3},b_{3}$) and ($x_{0}^{4},y_{0}^{4},r_{4},b_{4}$).1:  $x_{0}=\left(\sum_{i=1}^{64}k\left[i\right]\times 2^{i-1}\right)/2^{64}$;2:  $y_{0}=\left(\sum_{i=65}^{128}k\left[i\right]\times 2^{i-65}\right)/2^{64}$;3:  $r=\left(\sum_{i=129}^{192}k\left[i\right]\times 2^{i-129}\right)/2^{64}$;4:  $b=\left(\sum_{i=193}^{256}k\left[i\right]\times 2^{i-157}\right)/2^{64}$;5:     **for**
*j* = 1 to 4 **do**6:        $w_{j}=\left(\sum_{i=193+64j}^{256+64j}k\left[i\right]\times 2^{i-193-64j}\right)/2^{64}$;7:     **end**8:     **for**
*i* = 1 to 4 **do**9:        $x_{0}^{i}=\;\mathrm{mod}\;\left(x_{0}+w_{i}\times {\mathrm{sum}}_{1},1\right)$;10:       $y_{0}^{i}=\;\mathrm{mod}\;\left(y_{0}+w_{i}\times {\mathrm{sum}}_{2},1\right)$;11:       $r_{i}=\;\mathrm{mod}\;\left(r_{0}+w_{i}\times {\mathrm{sum}}_{3},6\right)$;12:       $b_{i}=\;\mathrm{mod}\;\left(b_{0}+w_{i}\times \mathrm{sum},1\right)$;13:    **end**

### 4.2. Image Preprocessing

Step 1: Scan three gray images to generate matrices ***P*${}_{1}$**, ***P*${}_{2}$**, ***P*${}_{3}$**, $M_{i}$ and $N_{i}$, which are rows and columns of the image matrix.

Step 2: If the image size is different, let $M_{max}$ = max($M_{i}$), $N_{max}$ = max($N_{i}$), where max() represents the maximum.

Step 3: The initial states $x_{0}^{1}$, $y_{0}^{1}$, $r_{1}$, $b_{1}$ generated in Algorithm 1 are substituted into LHCM for iteration. Then, the chaotic sequence generated by LHCM is added around the image with a small size so that the size of all three gray images is $M_{max}$ × $N_{max}$.

Step 4: Perform the channel fusion operation on the three gray images. The fused image is divided into blocks with the size of $L_{1}$ × $L_{2}$. The number of blocks can be obtained from Equation (Equation 10). Where *T* denotes the number of blocks, $L_{1}$ represents the length of the block divisible by $M_{max}$, and $L_{2}$ means the width of the block divisible by $N_{max}$.
(10)$$T=\left(M_{max}/L_{1}\right)\;\times \;\left(N_{max}/L_{2}\right)$$

### 4.3. Double Scrambling

Step 1: Substitute the initial state $x_{0}^{1}$, $y_{0}^{1}$, $r_{1}$, $b_{1}$ into the HSCM for (500 + *T*) iterations. The first 500 iterations are discarded to reach the full chaotic state, and two chaotic sequences $A_{1}$ and $A_{2}$ are obtained. The chaotic sequences $A_{1}$ and $A_{2}$ are processed by Equation (Equation 11) to obtain pseudo-random sequences $A_{1}^{*}$ and $A_{2}^{*}$.
(11)$$\left\{\begin{array}{c} A_{1}^{*}=\;\mathrm{mod}\;\left(\mathrm{round}\left(A_{1}\right)\times 10^{7},6\right)\\ A_{2}^{*}=\;\mathrm{mod}\;\left(\mathrm{round}\left(A_{2}\right)\times 10^{7},5\right) \end{array}\right.$$
where mod() is the modulo operation and round() denotes rounding a number to a specified number of digits.

Step 2: Firstly, according to the elements in $A_{1}^{*}$ find the corresponding rules in Table 1 for cross-plane scrambling between blocks. Then, the elements in $A_{2}^{*}$ correspond to the rules of Table 2 one by one, according to which, the block matrix is rotated and flipped. Finally, the three gray images are combined into a color image ***P*** ($M_{max}$ × $N_{max}$ × 3), which is shown in Figure 4.

Step 3: The initial state $x_{0}^{3}$, $y_{0}^{3}$, $r_{3}$, $b_{3}$ is substituted into HSCM to iterate (*L* + $L_{1}$) times and (*L* + $L_{2}$) times, respectively, and the previous *L* results are discarded to reach the full chaotic state. The remaining $L_{1}$ and $L_{2}$ results are stored in arrays ***A*${}_{4}$** and ***A*${}_{5}$**, respectively.

Step 4: Sort the arrays ***A*${}_{4}$** and ***A*${}_{5}$** separately to get the arrays ***A*${}_{4}^{\prime }$** and ***A*${}_{5}^{\prime }$**. The new arrays ***A*${}_{4}^{*}$** and ***A*${}_{5}^{*}$** record the position of each element in ***A*${}_{4}^{\prime }$** and ***A*${}_{5}^{\prime }$** in ***A*${}_{4}$** and ***A*${}_{5}$**, respectively.

Step 5: Think of each element in ***A*${}_{4}^{*}$** as the *x*-coordinate and each element in ***A*${}_{5}^{*}$** as the *y*-coordinate for inter-block scrambling. The scrambled image ***B*** is obtained by the following Equation (Equation 12):(12)$$B\left(i,j\right)=P\left(A_{4}^{*}\left(i\right),A_{5}^{*}\left(j\right)\right)$$

### 4.4. Bit-Level Grouping Diffusion

Because the bit-level operation can achieve a better scrambling and diffusion effect, the scrambling matrix ***B*** is converted into a binary matrix ***B*${}_{1}$**. However, bit-level algorithms need to process eight times more data than pixel-based algorithms. And single diffusion algorithm, especially the diffusion algorithm with XOR operation, is vulnerable to various attacks. Therefore, in this paper, matrix ***B*${}_{1}$** is uniformly divided into four parts, namely, 0, 1, 2, and 3, while ***B*${}_{1}$** is transformed into a bit-level matrix. When the pixel value of the image is converted to the bit-level matrix, each pixel value is converted to an 8-bit binary value. According to $2^{i}$, it can be divided into 8-bit group, 4-bit group, 2-bit group, and 1-bit group. At this time, the value of *i* corresponds to the value represented by the four parts. According to the grouping characteristics, each part’s diffusion process is realized by using bit spiral transformation, perfect shuffle, DNA encoding and V-shaped diffusion to improve the security of the encryption algorithm.

#### 4.4.1. Definition of Bit Spiral Transformation

When *i* = 0, $2^{i}$ = 1, the 8-bit binary number is regarded as a whole. At this time, the whole bit spiral transformation is performed on the 0th part of the bit-level matrix ***B*${}_{1}$**. The specific process is shown in Figure 5.

Step 1: Read the first row of pixels and put the read pixels into the newly generated matrix.

Step 2: Delete the elements of the first row of the original matrix and rotate the deleted matrix clockwise by $90^{\circ }$.

Step 3: Repeat steps 1 and 2; when the 0th part of the bit-level matrix ***B*${}_{1}$** has no elements, the diffusion matrix ***B*${}_{1}^{0}$** is obtained.

#### 4.4.2. Perfect Shuffle

When *i* = 1, $2^{i}$ = 2, the 8-bit binary number is divided into two parts, which can be used to perform bit diffusion on the first part of ***B*${}_{1}$** using the perfect shuffle algorithm.

Step 1: Divide the 8-bit binary sequence into two groups, and the sequence after grouping is $\left\{a_{1},a_{2},a_{3},a_{4},b_{1},b_{2},b_{3},b_{4}\right\}$.

Step 2: From the initial sequence $\left\{a_{1},a_{2},a_{3},a_{4},b_{1},b_{2},b_{3},b_{4}\right\}$ to the final sequence $\left\{b_{1},a_{1},b_{2},a_{2},b_{3},a_{3},b_{4},a_{4}\right\}$ by observing its subscript value, two closed cycles 1→2→4→8→7→5→1 and 3→6→3 can be formed.

Step 3: There is no intersection between two cycles, and the circle algorithm is executed for each cycle.

Step 4: The sequence after diffusion is $\left\{b_{1},a_{1},b_{2},a_{2},b_{3},a_{3},b_{4},a_{4}\right\}$.

Step 5: Repeat the above steps and finally obtain the diffusion matrix $B_{1}^{1}$.

#### 4.4.3. DNA Encoding

When *i* = 2, $2^{i}$ = 4, the 8-bit binary number is divided into four parts corresponding to the DNA sequence’s four bases A, G, C and T. The DNA algorithm can be used to perform the diffusion operation on the second part of ***B*${}_{1}$**.

Firstly, the DNA sequence is obtained by DNA encoding according to the DNA encoding rule 6 shown in Table 3. Then, the DNA sequence is decoded by the DNA encoding rule 2, shown in Table 3. Finally, the diffusion matrix ***B*${}_{1}^{2}$** is obtained.

#### 4.4.4. Definition of V-Shaped Diffusion

When *i* = 3, $2^{i}$ = 8, the 8-bit binary number can be divided into eight parts. At this time, the V-shaped diffusion operation is performed on the third part of ***B*${}_{1}$**.

Step 1: The initial state $x_{0}^{4}$, $y_{0}^{4}$, $r_{4}$, $b_{4}$ is substituted into HICM to iterate (1000 + *T*) times and discarded the results of the first 1000 iterations to generate the chaotic sequence $B^{\prime }$. The chaotic sequence $B^{\prime }$ is processed by Equation (Equation 13) to obtain the pseudo-random sequence $B^{*}$.
(13)$$B^{*}=\;\mathrm{mod}\;\left(\mathrm{round}\left(B^{\prime }\right)\times 10^{7},6\right)$$

Step 2: Assuming that each element in $B^{*}$ is $B_{j}$, separate a row from each $B_{j}$ bit according to the ‘V’ shape.

Step 3: Starting from the first row, read each row character in sequence, and then connect each column character together, and finally, obtain the ciphertext. The specific operation is shown in Figure 6.

Step 4: Splicing ***B*${}_{1}^{0}$**, ***B*${}_{1}^{1}$**, ***B*${}_{1}^{2}$**, ***B*${}_{1}^{3}$** to generate the primary diffusion matrix ***B*${}_{1}^{*}$**.

The process of bit-group-diffusion is shown in Figure 7.

### 4.5. Hexadecimal Addition and Subtraction Diffusion Operations

Step 1: Convert every four-bit binary number of the diffusion matrix ***B*${}_{1}^{*}$** into one hexadecimal digit to obtain the hexadecimal matrix ***C*${}^{1}$**.

Step 2: Formulate hexadecimal addition and subtraction rules such as: ‘A’ + ‘A’ = ‘4’ and ‘E’ − ‘3’ = ‘*B*’.

Step 3: The initial state $x_{0}^{1}$, $y_{0}^{1}$, $r_{1}$, $b_{1}$ generated in Algorithm 1 is substituted into HICM. Then, random matrices ***D*** and ***E*** of the same size as the scrambling matrix ***C*${}^{1}$** are generated in the same way as arrays *A*${}_{1}$ and *A*${}_{2}$. Finally, the ciphertext matrix ***C*** is obtained by two rounds of forward and reverse non-sequential diffusion by Equations (14) and (15).
(14)$$\left\{\begin{array}{c} F\left(1,1\right)=\;\mathrm{mod}\;\left(C^{1}\left(1,1\right)+D\left(1,1\right),256\right)\\ F\left(1,j\right)=\;\mathrm{mod}\;\left(C^{1}\left(1,j\right)+D\left(1,j\right)-F\left(1,j-1\right),256\right)\\ F\left(i,1\right)=\;\mathrm{mod}\;\left(C^{1}\left(i,1\right)+D\left(i,1\right)-F\left(i-1,1\right),256\right)\\ F\left(i,j\right)=\;\mathrm{mod}\;\left(C^{1}\left(i,j\right)+D\left(i,j\right)-F\left(i-1,j\right)-F\left(i,j-1\right),256\right) \end{array}\right.$$
(15)$$\left\{\begin{array}{c} C(M,N)=\;\mathrm{mod}\;(F(M,N)+E(M,N),256)\\ C(M,j)=\;\mathrm{mod}\;(F(M,j)+E(M,j)-C(M,j+1),256)\\ C(i,N)=\;\mathrm{mod}\;(F(i,N)+E(i,N)-C(i+1,N))\\ C(i,j)=\;\mathrm{mod}\;(F(i,j)+E(i,j)-C(i+1,j)-C(i,j+1),256) \end{array}\right.$$

### 4.6. Decryption Algorithm

The encryption process is reversible and the decryption process is an inverse process. Firstly, the key *k* is substituted into LHCM, HSCM and HICM maps to generate the chaotic sequence used in the encryption process. Then, the ciphertext image ***C*** is processed by inverse hexadecimal addition and subtraction diffusion operation to obtain the diffusion matrix ***B*${}_{1}^{*}$**, and the ***B*${}_{1}^{*}$** is proportionally divided into four sub-blocks and the scrambling matrix ***B*** is obtained by inverse packet diffusion. Finally, the scrambled image ***B*** is subjected to reverse coordinate matrix scrambling and reverse cross-plane scrambling to obtain the reconstructed image. The reconstructed image is separated to obtain three gray images.

## 5. Simulation Results and Security Analysis

According to the feature that a color image has three channels and a gray image has only one channel, this paper constitutes three gray images into a three-channel encrypted image. To verify the effectiveness and feasibility of the algorithm, three plaintext images of Boat, Lenna and Peppers of size 512 × 512 are selected for encryption and decryption experiments on MATLAB 2017 platform. The encryption and decryption effects are shown in Figure 8. Through the simulation results, it can be observed that the image encryption and decryption effect are good.

### 5.1. Key Space Analysis

In order to resist brute-force attacks, the key length of the image encryption algorithm based on chaos should be greater than 100 bits [34]. In this algorithm, the key mainly consists of two parts: (1) 256-bit initial key $k=\left\{x_{0},y_{0},r,b,w_{1},w_{2},w_{3},w_{4}\right\}$ generated by SHA-512. (2) DNA encoding and decoding rules. Since the encoding and decoding rules of DNA are two integers, the key space is $2^{256}+2>2^{100}$. Currently, the proposed algorithm has a large enough key space to resist brute-force attacks.

### 5.2. Key Sensitivity Analysis

Key sensitivity is an important indicator for testing the security of encryption algorithms. Key sensitivity can be tested in two ways. The first is to change the key to see if it can be decrypted correctly. The second is to change the key and observe the difference in ciphertext images. Experimentally, three plaintext images of Boat, Lenna and Peppers with 512 × 512 are selected for the key sensitivity test. In the trial, one group used the correct key for decryption. In the other two experiments, the initial value of $x_{0}$ was increased by $10^{14}$, and the value of the control parameter μ was reduced by $10^{14}$. The test results are shown in Figure 9a–c. As can be observed from the Figure 9b,c, after changing the keys $x_{0}$ and μ, it cannot be decrypted correctly. Similar results can be obtained if you change other keys for testing.

In order to obtain the ciphertext difference caused by the change of the key parameters more accurately, further experiments are carried out in this paper.The difference between decrypted images obtained by different keys (Figure 9a–c) is calculated, and the results are shown in Figure 9d–f. In Figure 9d–f, the ciphertext pixels obtained by the three groups of experiments are very different, and the image encryption algorithm based on CMCS has high key sensitivity.

The experimental results in Figure 9 show that the original image cannot be obtained even if the slightly changed key is used to decrypt the image. Only by using the correct key can the original image be decrypted correctly.

### 5.3. Histogram Analysis

The histogram directly demonstrates the distribution of pixel intensity. The more uniform the histogram of the image, the stronger the ability to resist statistical analysis attacks, and the more difficult the attacker to obtain image information. The algorithm first uses the double scrambling mechanism to change the pixel value positions of the image, but it cannot change the pixel values and make them uniformly distributed. Therefore, the pixel values are changed by bit grouping diffusion, hexadecimal addition and subtraction non-sequence diffusion. The histogram comparison before and after image encryption is shown in Figure 10.

From Figure 10, we can observe that the pixel values of three plaintext images are randomly distributed, and the pixel values of ciphertext images are uniformly distributed in the range of 0–255. The statistical properties of images are fundamentally changed. Therefore, this algorithm can effectively resist attacks based on statistical analysis.

### 5.4. Correlation Analysis

The strong correlation between adjacent pixels makes the image vulnerable to statistical attacks. Thus, breaking the strong correlation between adjacent pixels becomes one of the main purposes of encryption. The algorithm uses chaotic sequences generated by the CMCS system to perform an inter-block cross-plane scramble and intra-block coordinate matrix scramble of plaintext pixel values to eliminate the correlation between adjacent pixels.

Experimentally, 1000 pixel points are randomly selected from each of the image’s horizontal, vertical and diagonal directions for correlation analysis. The pixel value distribution of the plaintext image is shown in Figure 11, and that of the ciphertext image is shown in Figure 12. It can be observed that the correlation between adjacent pixels of ciphertext images is significantly reduced.

In addition, Table 4 shows the correlation data between the original and encrypted images of Lena images in the algorithm of this paper and other recent encryption algorithms. The closer the image correlation is to 0, the closer the image is to the random image. From Table 4, it can be observed that the correlation of adjacent pixels in horizontal, vertical and diagonal directions of the original image is close to 1, and the correlation of the encrypted image is close to 0. The results of comparing the image correlation under the action of different algorithms also proves that the algorithm in this paper has a better encryption effect.

### 5.5. Information Entropy Analysis

Information entropy reflects the randomness of information distribution in images. The higher the information entropy, the more random is the image. Assuming that the information in an image contains *L* different values, and the set of values is $\left(S_{0},S_{1}\ldots S_{N-1}\right)$, its information entropy H(S) can be expressed by the following Equation (Equation 16):(16)$$H\left(S\right)=-\sum_{i=0}^{L-1}P\left(S_{i}\right)\log\left[P\left(S_{i}\right)\right]$$
where $P(S_{i})$ represents the probability of $S_{i}$ appearing in the image ***S***. Obviously, for an 8-bit grayscale image, its ideal value H(s) = 8 [38]. The information entropy of the encrypted image of the proposed algorithm is the mean value of three channels. The information entropy of the plaintext image is shown in Table 5, and the comparison results between the proposed algorithm and other algorithms are shown in Table 6.

It can be observed from Table 5 that the information entropy of the ciphertext image is greatly improved compared with the plaintext image, which is close to 8, indicating that the ciphertext image is close to the random image. The results in Table 6 show that the proposed algorithm has more tremendous advantages than other algorithms.

### 5.6. Anti-Cropping Attack Analysis

As a common attack method, the cropping attack can be used to measure the robustness of an encryption algorithm. It crops against the ciphertext image and measures the algorithm’s robustness by deciphering the decrypted plaintext information. The experiment crops an arbitrary size region of the encrypted image at random and decrypts the cropped image, as shown in Figure 13.

From the perspective of decryption, even if different shapes are cropped to varying positions of the ciphertext image, the information of the original image is basically restored after decryption. Therefore, the algorithm proposed in this paper has the ability to resist cropping attacks.

### 5.7. Anti-Noise Attack Analysis

When the encrypted image is attacked by noise, most of the information of the plaintext image can still be obtained after decryption, which indicates that the algorithm has good robustness. Experiments add 5%, 10%, 20% salt and pepper noise to the plaintext image. Their corresponding decrypted images are shown in Figure 14. It can be observed from Figure 14 that even if the images are added with different proportions of noise, the basic information of the images can still be obtained after decryption, and the algorithm can resist noise attacks.

### 5.8. Analysis of Anti-Differential Attack

Differential attack means the attacker uses the same encryption algorithm to encrypt different plaintext images with similar pixels and determines the correlation between plaintext and ciphertext through the changes between additional encrypted images to crack the image encryption algorithm. The algorithm needs good diffusion characteristics to resist differential attacks. Therefore, the algorithm proposed in this paper first conducts block diffusion through bit spiral transformation, perfect shuffle, DNA encoding, and V-shaped diffusion. Then, the addition and subtraction algorithm of hexadecimal is used to conduct overall diffusion to resist differential attacks. The ability of the algorithm to resist differential attack is quantified by calculating the number of pixels’ change rate (NPCR) and unified averaged changed intensity (UACI). The calculation formula is as follows in Equations (17) and (18).
(17)$$\mathrm{NPCR}=\frac{1}{M\times N}\sum_{i=1}^{M}\sum_{j=1}^{N}Q\left(i,j\right)\times 100\%$$
(18)$$\mathrm{UACI}=\frac{1}{M\times N}\sum_{i=1}^{M}\sum_{j=1}^{N}\frac{\left|C_{1}\left(i,j\right)-C_{2}\left(i,j\right)\right|}{255}\times 100\%$$
where *M* and *N*, respectively, denote the length and width of the image, $C_{1}\left(i,j\right)$ and $C_{2}\left(i,j\right)$, respectively, represent the pixel values of the (i,j) point in the corresponding image of two plaintext images, Q(i,j) represents the different pixels of two ciphertext images.

For 8-bit gray images, the ideal expectation values for NPCR and UACI are 99.61% and 33.46%, respectively [42]. By changing one of the pixel values, the results of NPCR and UACI of the proposed algorithm in this paper compared with other literature algorithms for image Lena are shown in Table 7. It can be concluded from the table that the algorithm has a high ability to resist differential attacks.

### 5.9. NIST Test

NIST test consists of 15 subtests, which can be used to estimate the randomness of the sequence. The test results mainly demonstrate the advantages and disadvantages of a random sequence by analyzing the uniformity, consistency and pass rate of the sequence. The probability value (*p*-value) represents the uniformity of the sequence, and the proportional value represents the pass rate of the sequence. Each test gives a significant level of α = 0.01. If the *p*-value is more incredible than α, then the sequence has good randomness. Otherwise, the randomness of the sequence is insufficient.

The wash-ir (2250 × 2250 × 3) was selected as the test image for encryption. The first 8,000,000 pixels were converted to 64,000,000 binary sequences for the NIST test. The test results are shown in Table 8. Table 8 shows that the values of *p*-value are above 0.01, and the Proportion values are above 98%. The statistical test results show that the ciphertext sequence has good randomness.

### 5.10. Analysis of Encryption Efficiency

Encryption efficiency can be visually displayed by the running time of the image encryption algorithm. The encryption time of this algorithm is compared with other encryption algorithms, as shown in Table 9.

## 6. Conclusions

Analyzing the problems in the traditional scrambling-diffusion process cannot effectively balance the algorithm time and security. In this paper, we propose an image encryption algorithm based on a cascaded modulated chaotic system and block-scrambling-diffusion. In the encryption process, scrambling and diffusion are associated with plaintext, which can improve the algorithm’s security to a certain extent. Meanwhile, the design of block-scrambling-diffusion can effectively reduce the time cost. The experimental results demonstrate that the algorithm has ample key space, and can resist statistical analysis, noise attack, cropping attack and differential attack. However, the algorithm can only encrypt three gray images of different types into one ciphertext image and cannot encrypt multiple plaintext images into multiple ciphertext images in one encryption. In order to expand the scope of applicability of the algorithm, the next step will be to study the image encryption algorithm with an unlimited number and types of encryption.

## Figures and Tables

**Figure 1 entropy-24-01053-f001:**
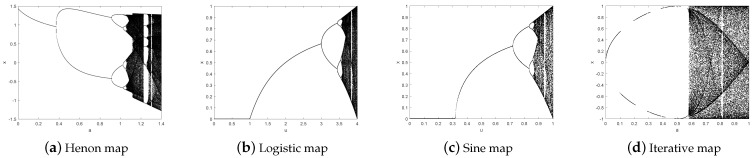
Chaotic map bifurcation graph.

**Figure 2 entropy-24-01053-f002:**
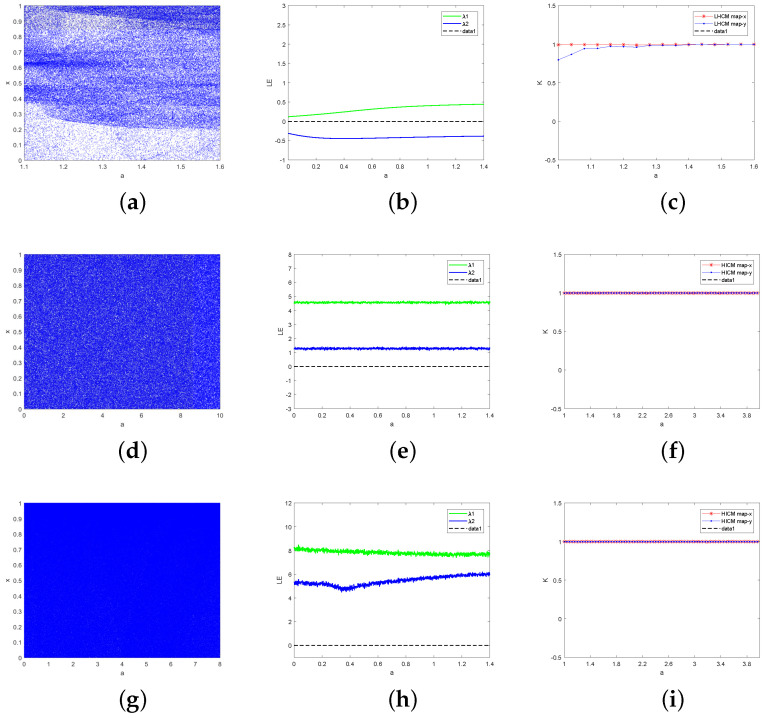
Performance analysis of chaotic map: (**a**) Bifurcation diagram of LHCM, (**b**) LE of LHCM, (**c**) 0–1 test of LHCM, (**d**) Bifurcation diagram of HSCM, (**e**) LE of HSCM, (**f**) 0–1 test of HSCM, (**g**) Bifurcation diagram of HICM, (**h**) LE of HICM, and (**i**) 0–1 test of HICM.

**Figure 3 entropy-24-01053-f003:**
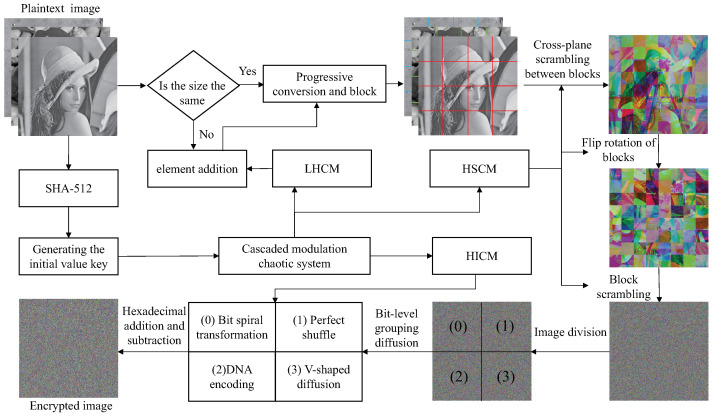
Encryption algorithm process diagram.

**Figure 4 entropy-24-01053-f004:**
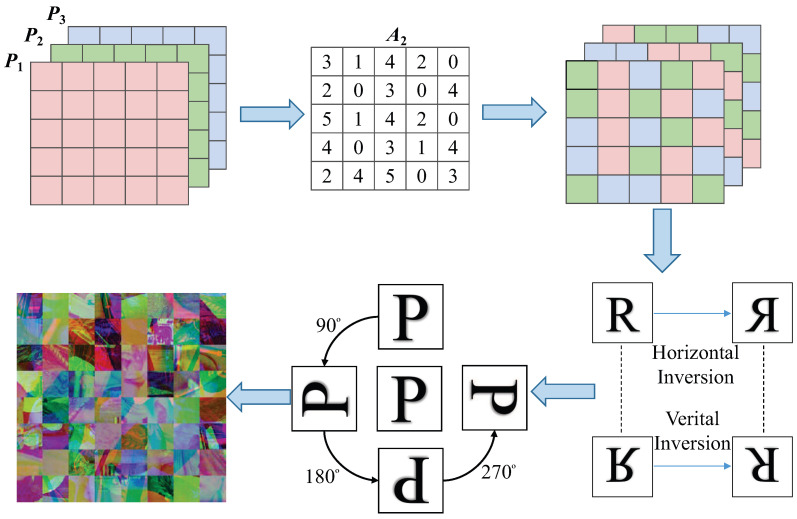
Multiple images are scrambled across planes.

**Figure 5 entropy-24-01053-f005:**
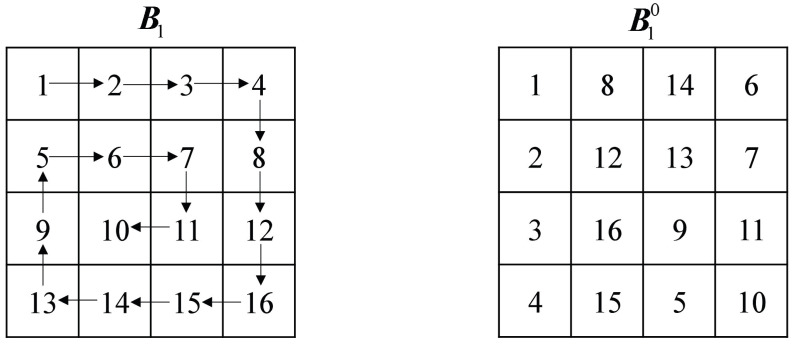
Bit spiral transformation.

**Figure 6 entropy-24-01053-f006:**
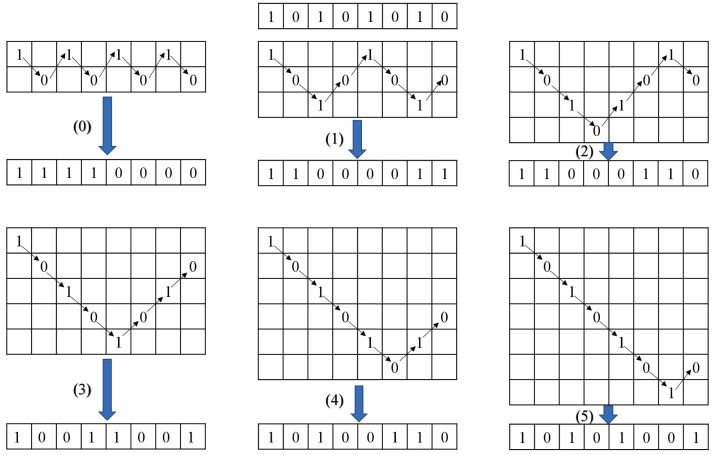
V-shaped diffuse.

**Figure 7 entropy-24-01053-f007:**
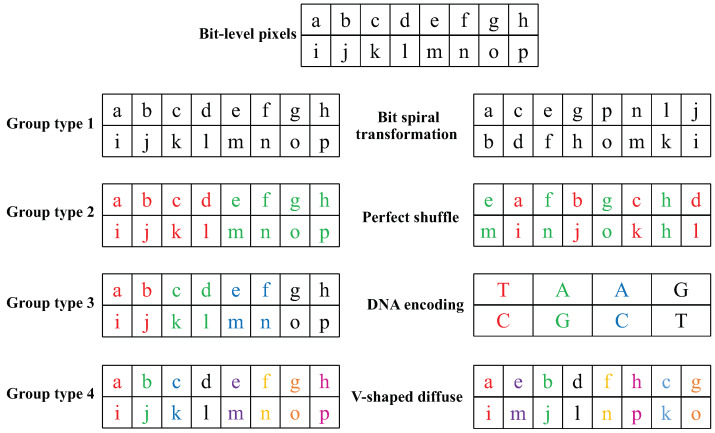
Bit-group diffusion.

**Figure 8 entropy-24-01053-f008:**
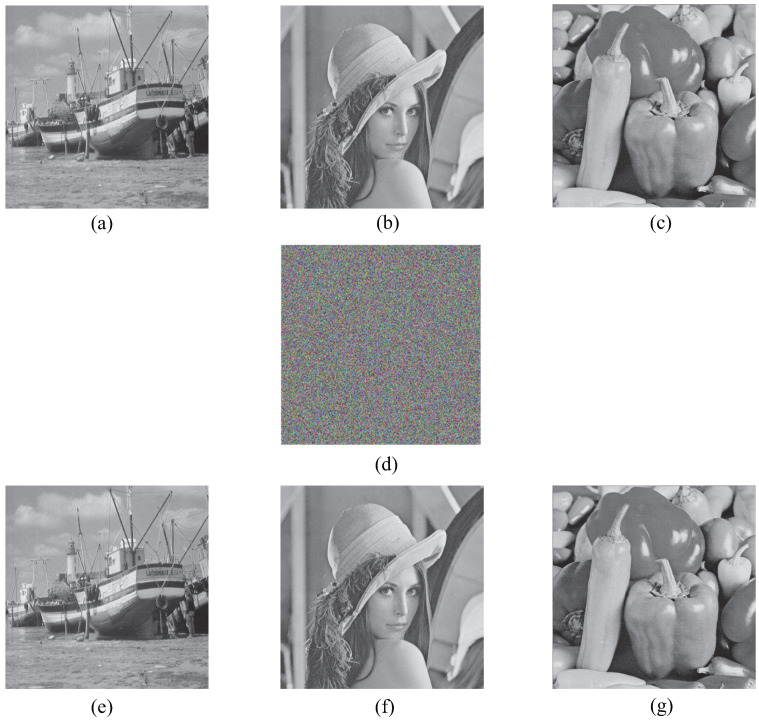
Encryption and decryption results: (**a**) Boat; (**b**) Lena; (**c**) Peppers; (**d**) Ciphertext image; (**e**) Decrypt image Boat; (**f**) Decrypt image Lena; (**g**) Decrypt image Peppers.

**Figure 9 entropy-24-01053-f009:**
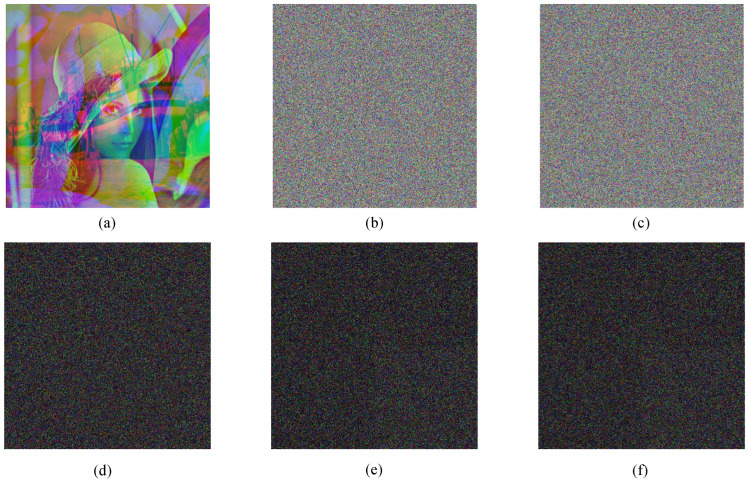
Key sensitivity test: (**a**) Combined image decrypted with correct key; (**b**) $x_{0}$ = $x_{0}$ + $10^{14}$; (**c**) μ = μ + $10^{14}$; (**d**) Difference image between (**a**) and (**b**), (**e**) Difference image between (**a**) and (**c**); (**f**) Difference image between (**b**) and (**c**).

**Figure 10 entropy-24-01053-f010:**
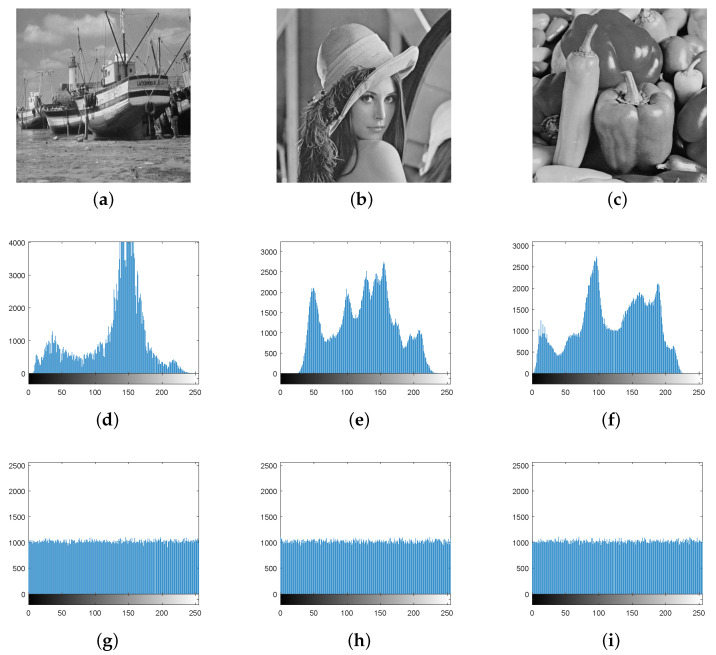
Histogram: (**a**) Boat, (**b**) Lena, (**c**) Peppers, (**d**) Boat plaintext histogram, (**e**) Lena plaintext histogram, (**f**) Peppers plaintext histogram, (**g**) R-channel ciphertext histogram, (**h**) G-channel ciphertext histogram, and (**i**) B-channel ciphertext histogram.

**Figure 11 entropy-24-01053-f011:**
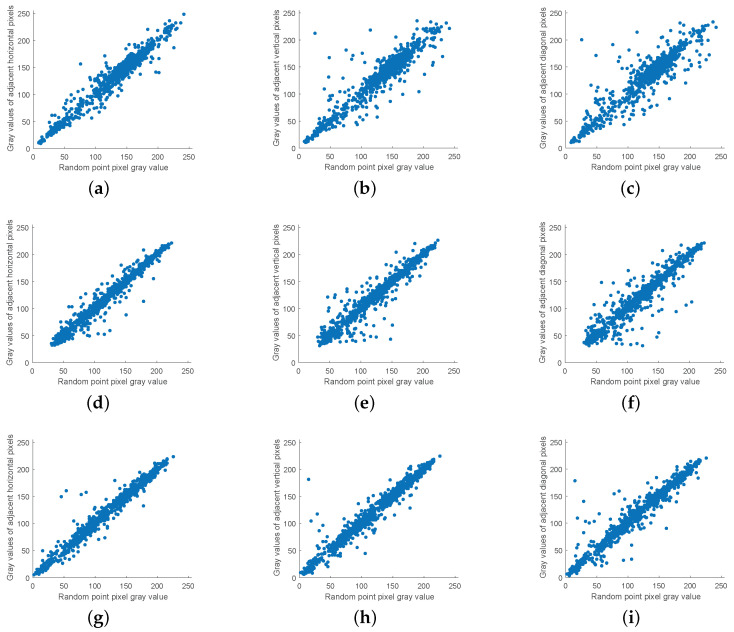
Plaintext image correlation analysis: (**a**) Boat horizontal direction, (**b**) Boat vertical direction, (**c**) Boat diagonal direction, (**d**) Lena horizontal direction, (**e**) Lena vertical direction, (**f**) Lena diagonal direction, (**g**) Peppers horizontal direction, (**h**) Peppers vertical direction, and (**i**) Peppers diagonal direction.

**Figure 12 entropy-24-01053-f012:**
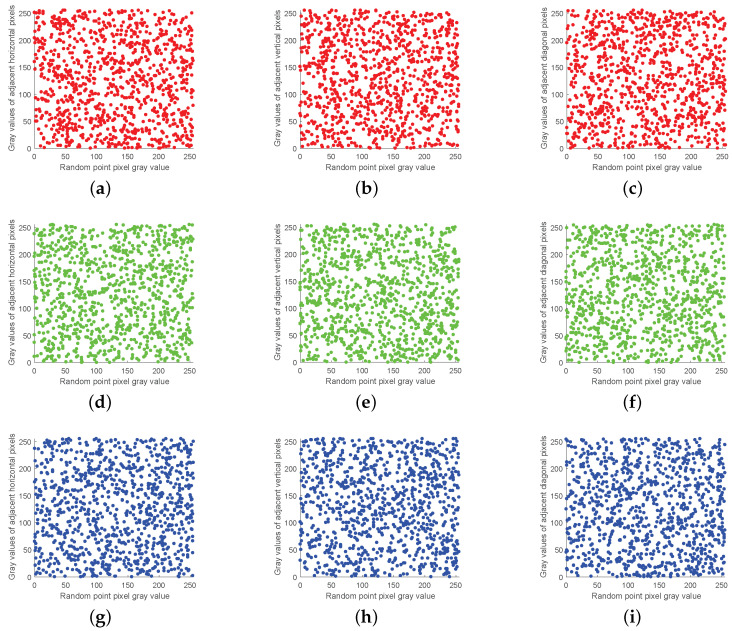
Ciphertext image correlation analysis: (**a**) R-channel ciphertext horizontal direction, (**b**) R-channel ciphertext vertical direction, (**c**) R-channel ciphertext diagonal direction, (**d**) G-channel ciphertext horizontal direction, (**e**) G-channel ciphertext vertical direction, (**f**) G-channel ciphertext diagonal direction, (**g**) B-channel ciphertext horizontal direction, (**h**) B-channel ciphertext vertical direction, and (**i**) B-channel ciphertext diagonal direction.

**Figure 13 entropy-24-01053-f013:**
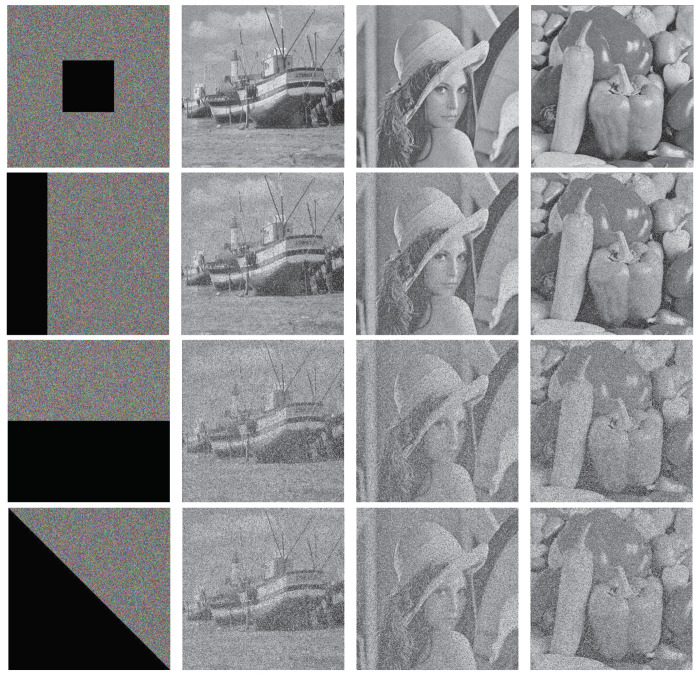
Anti-cropping attack analysis.

**Figure 14 entropy-24-01053-f014:**
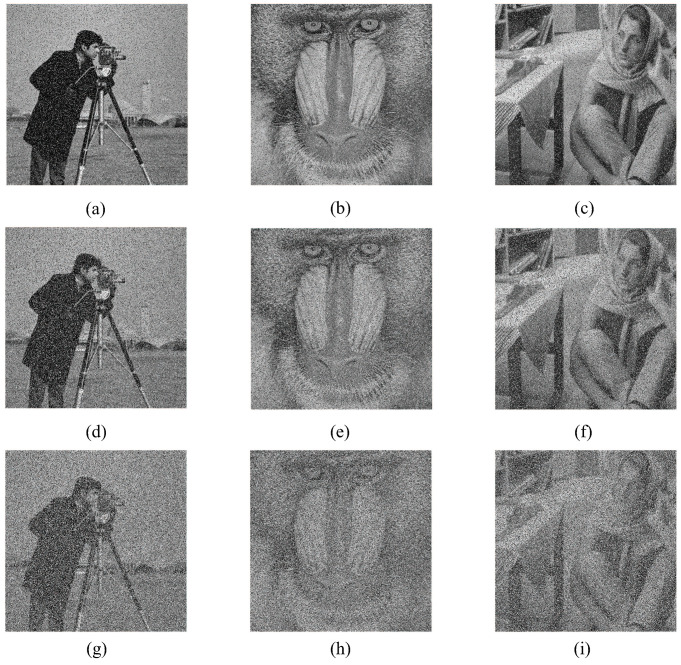
Anti-noise attack analysis: (**a**) 5% pepper noise; (**b**) 5% pepper noise; (**c**) 5% pepper noise; (**d**) 10% pepper noise; (**e**) 10% pepper noise; (**f**) 10% pepper noise; (**g**) 20% pepper noise; (**h**) 20% pepper noise; (**i**) 20% pepper noise.

**Table 1 entropy-24-01053-t001:** Cross plane scrambling rules.

0	1	2	3	4	5
$P_{1}$	$P_{1}$	$P_{2}$	$P_{2}$	$P_{3}$	$P_{3}$
$P_{2}$	$P_{3}$	$P_{1}$	$P_{3}$	$P_{1}$	$P_{2}$
$P_{3}$	$P_{2}$	$P_{3}$	$P_{1}$	$P_{2}$	$P_{1}$

**Table 2 entropy-24-01053-t002:** Rotation and inversion rules.

0	1	2	3	4
Rotate $90^{\circ }$	Rotate $180^{\circ }$	Rotate $270^{\circ }$	Horizontally Inversion	Vertically Inversion

**Table 3 entropy-24-01053-t003:** Eight kinds of DNA codes satisfying complementarity rules.

Rule	0	1	2	3	4	5	6	7
A	00	00	01	01	10	10	11	11
T	11	11	10	10	01	01	00	00
G	01	10	00	11	00	11	01	10
C	10	01	11	00	11	00	10	01

**Table 4 entropy-24-01053-t004:** Correlation coefficient.

Image	Horizontal Direction	Vertical Direction	Diagonal Direction
Boat	0.9450	0.9758	0.9283
Peppers	0.9732	0.9847	0.9550
Lena	0.9666	0.9823	0.9566
Ref. [35]	0.0032	−0.0182	−0.0021
Ref. [36]	0.0635	0.1981	0.1698
Ref. [37]	0.0041	0.0043	0.0084
**Our scheme**	**0.0020**	**−0.0006**	**−0.0062**

**Table 5 entropy-24-01053-t005:** Information entropy plain images and encrypted images.

Algorithm	Image	Information Entropy	
		Plain Image	Cipher Image
Our scheme	Boat	7.1914	**7.9998**
	Lena	7.4451	
	Peppers	7.5937	

**Table 6 entropy-24-01053-t006:** Information entropy of different algorithms.

Algorithm	Our Scheme	Ref. [39]	Ref. [40]	Ref. [41]	Ref. [34]
Information entropy	**7.9998**	7.9994	7.9996	7.9995	7.9994

**Table 7 entropy-24-01053-t007:** NPCR and UACI of different encryption algorithms.

Algorithm	NPCR	UACI
**Our scheme**	**99.6289**	**33.5006**
Ref. [39]	99.6250	33.4510
Ref. [40]	99.1841	33.5284
Ref. [41]	99.5907	33.4811
Ref. [34]	99.6208	33.5025

**Table 8 entropy-24-01053-t008:** NIST test.

Sub-Tests	*p*-Value	Proportion	Pass/Fail
Frequency Test	0.275709	62/64	Pass
Block Frequency Test	0.134686	64/64	Pass
Cumulative Sums	0.931952	62/64	Pass
Runs Test	0.706149	64/64	Pass
Longest Run Test	0.350485	64/64	Pass
Rank Test	0.568055	61/64	Pass
FFT	0.602458	63/64	Pass
Non-Overlapping Template Test	0.772760	64/64	Pass
Overlapping Template Test	0.213309	64/64	Pass
Universal Test	0.568055	63/64	Pass
Approximate Entropy Test	0.637119	63/64	Pass
Serial Test	0.437274	63/64	Pass
Random Excursions Test	0.585209	44/44	Pass
Random Excursions Variant Test	0.739918	44/44	Pass
Linear Complexity Test	0.299251	64/64	Pass

**Table 9 entropy-24-01053-t009:** Running time of encryption algorithm.

Algorithm	Image	Times(s)
**Our scheme**	**3 × 256 × 256**	**0.9375**
AES scheme [43]	1 × 256 × 256	5.5312
Ref. [44]	1 × 256 × 256	2.0422
Ref. [45]	3 × 256 × 256	1.5910
Ref. [46]	3 × 256 × 256	1.5700

## Data Availability

Not applicable.

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
