# Peer review of "Multi-Image Encryption Algorithm Based on Cascaded Modulation Chaotic System and Block-Scrambling-Diffusion"

_entropy, 2022, doi:10.3390/e24081053_

Round 1

Reviewer 1 Report

The goal of the paper is very unclear to me.
I think sources of such unclearness are that there are no formal definitions and no security notions
in the standard cryptographic paper.

*My understanding of the overview of the image encryptions is as follows.
(1) Plaintext to be encrypted is pictures P1,...,Pm (a set of matrices over integers, m=3 in this paper)
(2) Generate a short key k by a function (this paper employs SHA-512) and share it by a secure channel
(3) Ciphertext is ct=Enc(k,{P1,...,P3})
(4) Decryption is performed by a function Dec(k,ct)

Here, the construction of the encryption function is very puzzling from a lot of magic numbers and magic functions,
whereas an obfuscation of programming does not imply the security of output data.
So, I think the author should claim the encryption function is secure in some sense.
The security analyses in Section 5 claim that the output image looks like random noises, however,
it does not mean that recovering the plaintext from the ciphertext without using the legitimate key
is computationally hard.
In addition, the decryption algorithm is not given though the title of Section 4 is "Encryption and Decryption Algorithm."

*From the viewpoint of cryptographic theory, this is insecure because it does not have an indistinguishability property.
In general, if an encrypting function is deterministic (i.e., it doesn't use any random function,) the encryption scheme
is not secure in the sense of indistinguishability.
Also, since the key is computed deterministically, the same plaintext is always encrypted to the same ciphertext.

*From the viewpoint of computational costs, I doubt the advantage of the proposed scheme
to the standard way: compress the image by zip and then encrypt by AES.
(The advantage of the standard way is that it can hide the size of images.)

*The authors say "traditional encryption technologies such as DES and AES are no longer applicable due to ..."
but I think SHA-512 is typically slower than AES. At least they are comparable to each other.

*From the encrypting key, it generates a sequence of floating numbers via some "heavy" operations such as
sine function and multiplication by pi.
I think implementations of sine and pi can be different in many systems, and it produces small errors which
make big gaps (butterfly effect of chaotic maps) between the sender and receiver.

*The cross-plane scrambling looks very heavy because it requires copying of distant memory blocks.
This should be a disadvantage.

*On the used chaotic maps in Section. 2.
Did you check that they produce good random numbers for any initial states?

Author Response

Please see the attachment. Thank you for your review.

Reviewer 2 Report

This work presents a new multi-image encryption process using memristive map. The manuscript can be divided into two main parts. First 2-D map is designed using a discrete memristor model equation. The map is based on the well-known logistic map, sine map, and tent map. First, the performances of the designed map are presented using three classic tools like phase, bifurcation diagram, and corresponding Lyapunov exponents. The analysis shows that the map has hyperchaos in its dynamics. Second, the sequences of the designed map are exploited to construct a new protocol of information communication. The work seems interesting however the second part lacks a performance analysis of the communication protocol. 

The paper requires to be improved and here are some suggestions:

1. The motivation of the work is not stated.

2. More simulation results are required to bring more light to the performance of the proposed communication protocol.

3. NIST test must be investigated.

4. What is the main property of a discrete memristor and what makes this type of memristor better than a continuous-time memristor?

5. The analysis of the designed map is very poor. What is new in the dynamics of the proposed map regarding some recent achievements? A comparative analysis is required.

6. The main objectives of this work should be clearly stated in the introduction.

7. Complexity analysis is required.

8. Language must be improved.

Author Response

Dear Editor,

Thank you for allowing a resubmission of our manuscript, with an opportunity to address the reviewers’ comments.

First of all, I am very sorry to bother you because I find your comments seem to be inconsistent with the subject of our paper.

  1. We didn't use the memristive map, we propose cascade modulation chaotic systems (CMCS).
  2. The maps we proposed are not based on Logistic, Sine, and Tent maps. We propose a chaotic system that can generate multiple maps, not a single map. Based on Henon, Logistic, Sine and Iterative mappings, the maps generated by CMCS are illustrated.
  3. We didn't use the phase, bifurcation diagram, and corresponding Lyapunov exponents to verify the performance of the proposed chaotic map. But through the bifurcation diagram, Lyapunov exponents and 0-1 test to verify the performance of the proposed chaotic map.
  4. We tested NIST in our paper.

Reviewer 3 Report

The paper is well written, is clear and can be well understood. It is relevant and can be of help and interest for the information security research community.

 The topic is suitable for the journal, of broad international interest, significant and novel. The paper is clearly presented. The references are relevant, up to date, accessible, adequate and complete.

 Recently it has been demonstrated that the image encryption algorithms based on a single chaotic map have serious problems of security. In this paper a proposition of a cascade modulation chaotic system (CMCS) that can generate 2 multiple chaotic maps is presented. In the encryption process, scrambling and diffusion are associated with plaintext, which can improve the algorithm’s security to a certain extent. Meanwhile, the design of block-scrambling-diffusion can effectively reduce the time cost. The experimental results show that the algorithm has ample key space, and can resist statistical analysis, noise attack, cropping attack and differential attack.

 The work is enough relevant to be published by the journal Entropy, because the interest of the chaotic information encryption methods is very high for the security research environment.

Author Response

Thank you for your review and wish you a happy life.

Round 2

Reviewer 1 Report

For my naggy and too-detailed comments, the authors gently replied and modified their manuscript.
Although I'm keeping the obstinate position, I think the paper should be published because
it lights up advantages and disadvantages of image encryption.

*I think the fundamental problem to be discussed in this area is to make formal definitions like the public-key encryption
like https://www.cs.virginia.edu/~mohammad/courses/crypto/sp18/notes/Lecture_08.pdf,
not producing complicated/obfuscated/puzzled pseudorandom functions.

*I don't understand why the authors consider the key sensitivity since
the key is computed deterministically from input image in the first version of the scheme.
In what scenario would it be a problem?

*In response 5, the authors added to use a random bits k4 in key generation.
How do you share k4 between the sender and receiver with keeping security?
In order to discuss such problem, you need to fix the model of communication and formal definitions of
functions used in the scheme.
This is one of the reason that I wanted the formal definitions.

*I suspect the benchmark results in Table.9.
The authors says it takes 5.5 seconds to encrypt one 256*256 image by AES.
The size of 256*256 image should be 192 or 256 KB, assuming 24bits or 32bits per pixel respectively.
So, the encryption speed of your AES benchmark is less than 50KB/sec.
On the other hand, published benchmarking results such as
https://www.bearssl.org/speed.html
claims that 25-700 MB/sec for encrypting.
500 difference is not negligible.

Reviewer 2 Report

There are no other suggestions for the paper

Author Response

Dear Editor,

Thank you very much for your review, wish you a happy life.